# Hydrogel-Based Therapeutics for Pancreatic Ductal Adenocarcinoma Treatment

**DOI:** 10.3390/pharmaceutics15102421

**Published:** 2023-10-04

**Authors:** Jinlu Liu, Wenbi Wu, Qing Zhu, Hong Zhu

**Affiliations:** 1Division of Abdominal Tumor Multimodality Treatment, Cancer Center, West China Hospital, Sichuan University, Chengdu 610041, China; liujinlu@stu.scu.edu.cn (J.L.); newzhuqing1972@yahoo.com (Q.Z.); 2Department of Biotherapy, Cancer Center, State Key Laboratory of Biotherapy, West China Hospital, Sichuan University, Chengdu 610041, China; xiaojiujiugg@126.com

**Keywords:** pancreatic ductal adenocarcinoma, hydrogel, drug-delivery system, immunosuppressive tumor microenvironment

## Abstract

Pancreatic ductal adenocarcinoma (PDAC), one of the deadliest malignancies worldwide, is characteristic of the tumor microenvironments (TME) comprising numerous fibroblasts and immunosuppressive cells. Conventional therapies for PDAC are often restricted by limited drug delivery efficiency, immunosuppressive TME, and adverse effects. Thus, effective and safe therapeutics are urgently required for PDAC treatment. In recent years, hydrogels, with their excellent biocompatibility, high drug load capacity, and sustainable release profiles, have been developed as effective drug-delivery systems, offering potential therapeutic options for PDAC. This review summarizes the distinctive features of the immunosuppressive TME of PDAC and discusses the application of hydrogel-based therapies in PDAC, with a focus on how these hydrogels remodel the TME and deliver different types of cargoes in a controlled manner. Furthermore, we also discuss potential drug candidates and the challenges and prospects for hydrogel-based therapeutics for PDAC. By providing a comprehensive overview of hydrogel-based therapeutics for PDAC treatment, this review seeks to serve as a reference for researchers and clinicians involved in developing therapeutic strategies targeting the PDAC microenvironment.

## 1. Introduction

Pancreatic cancer, originating from pancreatic duct epithelial cells, is the most common cause of death from digestive system malignancies. It has been reported that, of 49,600 PDAC cases in 2020 worldwide, 46,600 people died (93.95% mortality rate) [1]. At the point of diagnosis, about 90% of patients are shown to have already progressed into the advanced stage and are not eligible for radical surgery to remove the tumor [2]. For these patients, oxaliplatin-based or gemcitabine-based chemotherapy are suggested as the first-line therapy regimen [3]. However, the survival of patients with advanced PDAC is less than 1 year [4]. Meanwhile, the severe side effects of systematic chemotherapy, including myelosuppression, organ dysfunction, etc., hinder patients from benefiting from the treatment [5]. Therefore, effective and safe therapeutics for PDAC treatment are urgently required.

The extracellular matrix (ECM) of PDAC is a typical desmoplastic environment which is rich in collagen with high stiffness, which acts as a natural barrier to the infiltration of therapeutics and immune cells [6,7]. Most stroma cells in the TME are cancer-related fibroblasts (CAFs), which can introduce a large amount of fiber and collagen into the TME. In some conditions, the number of these fibroblasts can exceed that of cancer cells [8,9]. CAFs in the PDAC TME can be generated by pancreatic stellate cells (PSCs) which are activated by transforming growth factor beta 1 (TGFβ1) or other mediators [10,11,12]. In the TME of PDAC, immunosuppressive stromal cells increase, such as regulatory T cells (Tregs), tumor-associated macrophages (TAMs), and myeloid-derived suppressor cells (MDSCs), while the antitumor lymphoid populations decrease, thus impairing the immune-mediated cytotoxic effects targeting at tumor cells [13,14,15]. Even the cytotoxic T cells infiltrating into the TME are lacking in the activation markers, such as granzyme B (GRZB) and Interferon-γ (IFNG) [16]. Hypoxia is another characteristic of the PDAC TME due to the extensive desmoplastic stoma and endows the tumor cells with more aggressiveness, increasing their survival and growth under hostile conditions [17]. Therefore, the PDAC TME is characterized as immunosuppressed, resembling a cold desert, and it poses challenges for immune cell infiltration due to its stiffness.

Compared to traditional chemotherapy, an effective drug-delivery system that delivers drugs directly to the tumor to achieve controlled and sustained drug release, avoiding the systemic circulation of chemotherapy drugs and greatly reducing systemic adverse reactions, is necessary for the treatment of PDAC [18]. As a versatile biomaterial, hydrogels have attracted considerable research attention for this purpose, showing great potential as scaffolds or substrates for drug delivery, cell incubation, and as implants [19]. Hydrogels are hydrophilic polymer networks formed by the physical and/or chemical crosslinking of polymer substances and have been widely used to deliver cargoes, like drugs, nucleic acid, protein, and cells [20,21]. The release of these cargoes from hydrogels is mainly affected by factors such as porosity, network expansion/degradation, size of molecules to be released, drug–polymer interactions, and environmental stimuli like pH, temperature, enzymes, etc. [22]. The ideal hydrogel would be able to deliver cargoes specifically into the targeted lesion and could release them in a slow and sustained manner. Recently, researchers have been working diligently to develop multiple hydrogel-based therapeutics for PDAC treatment. 

In this review, we will present a picture of the immunosuppressive PDAC TME, summarize the hydrogel-based therapeutics used in PDAC, categorizing them based on the type of cargo they deliver, and describe the remodeling of the TME resulting from these therapeutics (Figure 1). Furthermore, we will discuss the potential candidate drugs for use in hydrogel-based therapeutics for PDAC.

## 2. The Immunosuppressive TME of PDAC

The immunosuppressive microenvironment in PDAC is intricately associated with the presence of specific oncogenes which undergo mutations in the context of PDAC. The activation of downstream pathways by these oncogenes assumes a pivotal role in shaping the immunosuppressive characteristics of the TME [23,24]. Notably, KRAS mutations are highly prevalent in approximately 93% of PDAC individuals, with frequent occurrences of mutations such as G12D and G12C [25]. The presence of KRAS mutations amplifies the secretion of chemokines, specifically CCL2, CCL5, and CXCL8, which subsequently recruit immunosuppressive cells (including TAMs, MDSCs, and Tregs) to the TME. These immune cells are capable to impair the activation and functionality of effector immune cells like cytotoxic T lymphocytes (CTLs) and natural killer cells [24,26,27]. Consequently, these physiological processes collectively contribute to the emergence of an immunosuppressive microenvironment in PDAC.

Additionally, PDAC cells actively impede the immune response against tumor cells through various evasion tactics. A key mechanism involved in this evasion is the downregulation of major histocompatibility complex class I molecules (MHC I) on the tumor cell surface, which are essential for the CTL recognition [28]. Moreover, PDAC cells exhibit a significant upregulation of CD47, a cell surface protein which functions as a potent “do not eat me” signal. CD47 interacts with macrophage SIRPα receptors and natural killer cell SIRPγ receptors, effectively inhibiting phagocytosis of tumor cells through immune cells [29]. Meanwhile, PDAC cells generate indolamine 2,3-dioxygenase (IDO) to degrade tryptophan—the vital amino acid required for CTL survival and activation. Consequently, this mechanism ultimately leads to the initiation of apoptosis and the subsequent impairment of T-cell function [30]. Furthermore, PDAC cells downregulate the expression of human leukocyte antigen DR isotype (HLA-DR) and CD40. This downregulation directly inhibits effector CD8+ T cells through immature dendritic cells (DCs) [31,32]. Collectively, PDAC cells driven by oncogenes possess inherent immunosuppressive properties which disrupt normal host immune responses through multiple mechanisms. We will further explain the immunosuppressive cells in PDAC TME in the following section (Figure 2).

### 2.1. MDSCs

MDSCs represent a heterogeneous population of immune cells which serve a crucial role in the suppression of antitumor immune responses, especially in cases of PDAC. These cells are known to accumulate within both the TME and the peripheral blood, actively promoting immune suppression and significantly contributing to tumor progression [33,34,35]. MDSCs can be divided into two main categories: monocytic MDSCs (M-MDSCs) and polymorphonuclear MDSCs (PMN-MDSCs). These are also referred to as granulocytic MDSCs (G-MDSCs). PMN-MDSCs originate from immature neutrophils in the bone marrow, while M-MDSCs share phenotypic similarities with monocytes and exhibit a monocytic morphology. Notably, M-MDSCs possess the ability to differentiate into TAMs and DCs [36]. MDSCs originate from hematopoietic precursor cells (HSCs) and are recruited to the TME under the regulation of inflammatory mediators, such as IL-6, IL-10, IL-8, CCLs, CXCLs, GM-CSF, VEGF, and TNF-α, during the development and progression of PDAC [37,38]. The differentiation of HSCs into common myeloid progenitors (CMPs) occurs, which subsequently differentiate into immature myeloid cells (IMCs). Conversely, in the presence of tumor-derived cytokines, such as CXCLs and GM-CSF, the customary differentiation process of IMCs is perturbed, compelling their transition into MDSCs and subsequent expansion and activation within peripheral blood, bone marrow, or tumor lesions [39,40]. Studies conducted on the spontaneous PDAC model of the genetically engineered and mutant mice (GEMMs) have demonstrated a gradual increase in MDSCs in the TME as the tumor advances, a pattern also observed in the spleen and peripheral blood [34,35]. 

The mechanisms underlying immune suppression in PDAC involve distinct contributions from M-MDSCs and PMN-MDSCs [41,42]. The generation of ROS by PMN-MDSCs can result in the impairment of T-cell receptor signaling, consequently leading to reduced T-cell activation and compromised functionality. Furthermore, PMN-MDSCs possess the ability to induce T-cell apoptosis and restrict the availability of crucial nutrients such as cysteine, tryptophan, and arginine, required for T-cell survival and proliferation. Moreover, PMN-MDSCs can also enhance the immunosuppressive responses by promoting the expansion and activation of Tregs [43,44,45]. On the other hand, M-MDSCs promote Treg expansion by releasing cytokines such as IL-10 and TGF-β, creating an immunosuppressive environment which dampens antitumor immune responses [46,47]. Additionally, M-MDSCs inhibit T-cell function by secreting immunosuppressive factors such as arginase-1 and inducible nitric oxide synthase (iNOS). These enzymes deplete essential amino acids, such as arginine and produce nitric oxide, impairing T-cell proliferation and functionality [48]. MDSCs also contribute to the cancer progression by promoting tumor angiogenesis, tumor cell invasion, and metastasis to distant organs [34,49,50]. In PDAC, gemcitabine (GEM)-resistant pancreatic cancer cells induce the recruitment of MDSCs into the TME through GM-CSF secretion, leading to the inhibition of CD8+ T-cell proliferation and resistance to GEM. Additionally, increased MDSCs in the tumor tissue of patients with PDAC after radiotherapy are associated with reduced treatment efficacy [51].

### 2.2. TAMs

TAMs are derived either from monocytes infiltrated within the TME or preexisting tissue-resident macrophages [52]. In the initial stage of tumor invasion, tissue-resident macrophages primarily constitute TAMs; meanwhile, later, a substantial proportion of TAMs are derived from infiltrating monocytes. However, in established mouse PDAC models, TAMs primarily consist of monocyte-derived macrophages [53]. Functionally, TAMs can be classified into two distinct phenotypes: M1 TAMs, which exhibit antitumor activity; M2 TAMs, which exert protumor and immunosuppressive properties. M1 TAMs function as antigen-presenting cells (APCs), expressing Interleukin 12 (IL12) and tumor necrosis factor (TNF) to combat tumor cells. In contrast, M2 TAMs produce cytokines such as Interleukin 10 (IL-10), TGF-β, and arginase, which inhibit T-cell cytotoxicity and proliferation. Additionally, M2 TAMs demonstrate reduced major histocompatibility complex class II (MHC-II) expression, thereby weakening their antigen presentation ability [54,55]. The recruitment of monocytes to the TME and their subsequent differentiation into macrophages are orchestrated by the overproduction of colony-stimulating factor 1 (CSF1) and chemokine ligand 2 (CCL2) within the TME [56,57].

### 2.3. Tregs

The high infiltration of Tregs within the PDAC TME is strongly associated with an unfavorable prognosis for patients [58]. Cytokines secreted by PDAC cells, such as CCL5, TGF-β, and IL-10, play a crucial role in the recruitment and accumulation of Tregs into the TME. The presence of Tregs in the TME exerts inhibitory effects on the antitumor immune response, thereby contributing to tumor progression across various cancer types [57,59]. Tregs are capable of producing inhibitory cytokines, including IL-10, TGF-β, and IL-35, which imped the function of effector cells. Additionally, Tregs promote the generation of adenosine (AMP) within the TME through the expression of extracellular enzymes CD39 and CD73, resulting in the suppressive and anti-proliferative effects of effector cells. Nevertheless, the precise role of Tregs in the context of PDAC remains a subject of controversy. In the transplantation model of PDAC, depletion of Foxp3+ Treg cells significantly increased the infiltration of CD8+ T cells within the TME [60]. Conversely, an opposing outcome was reported by Zhang, Y.Q. et al., in which the depletion of Foxp3+ Treg cells accelerated tumor progression in the GEMM model of PDAC, potentially due to the subsequent increase in MDSC resulting from chemokine release following Treg reduction. The investigators hypothesized that these conflicting observations may be attributed to the two different stages of tumor progression, as represented by the two distinct tumor models, with the transplantation model corresponding to a more advanced stage of the PDAC compared to the GEMM model [15]. 

### 2.4. CAFs

CAFs are key cellular components of the TME in PDAC and have a profound impact on tumor progression and therapy resistance. One significant mechanism by which CAFs contribute to therapy resistance is through remodeling the ECM and creating a physical barrier which restricts the penetration of therapeutic agents and immune cells into the TME [9,61]. The transformation of CAFs is primarily initiated by the stimulation of intrinsic fibroblasts or stellate cells within the tissues, often mediated by growth factors such as TGF-β [62,63]. Following activation, CAFs secrete a wide range of cytokines which play critical roles in tumor development and immune modulation. For instance, vascular endothelial growth factor (VEGF) is involved in the regulation of tumor angiogenesis and the formation of an aberrant vascular network which sustains tumor growth [64]. Interleukin 6 (IL-6), another cytokine secreted by CAFs, promotes MDSC differentiation while concurrently inhibiting cytotoxic T cells [65,66]. Furthermore, CAFs can induce an epithelial–mesenchymal transition in the neighboring epithelial cells, a process associated with enhanced invasive and metastatic potential [67]. To classify CAFs and better understand their heterogeneity, researchers have identified specific markers such as fibroblast activating protein (FAP), α-smooth actin (α-SMA), fibroblast-specific protein 1 (FSP1), and platelet-derived growth factor receptor (PDGFR) [68]. FAP, in particular, is specifically expressed in CAFs and has been explored as a target for therapeutic interventions and imaging diagnosis in the context of CAFs. Recent advancements in single-cell sequencing technology have allowed researchers to gain deeper insights into the functional heterogeneity of CAFs [69]. This approach has revealed that CAFs can adopt a pro-inflammatory phenotype, thus promoting cancer cell growth, enabling immune evasion and facilitating metastatic dissemination. This is achieved through the secretion of cytokines and chemokines which modulate the immune environment in the TME. These factors recruit inhibitory myeloid and Treg cells, impair the function of cytotoxic lymphocytes and dendritic cells, and promote the polarization of M2 and type 2 helper T (Th2) cells, all of which contribute to immunosuppression and tumor progression. 

## 3. Hydrogel-Based Therapeutics for PDAC

Hydrogels are three-dimensional networks of hydrophilic polymers and are promising candidates for drug delivery due to their unique properties. In recent years, the involvement of hydrogels in PDAC treatment has gained attention due to their potential therapeutic benefits [70,71]. Based on the type of monomers, hydrogels applied for PDAC treatment can be classified into three categories, including synthetic hydrogels, natural hydrogels, and semi-synthetic hydrogels. Synthetic hydrogels are composed of synthetic polymers such as polyethylene glycol (PEG) [72], polyacrylic acid (PAA) [73], polyvinyl alcohol (PVA) [74], and polylactic acid (PLA) (Figure 3a) [75], which are tunable, reproducible, and stable; however, they might be limited by their poor biocompatibility and biodegradability [76]. Meanwhile, natural hydrogels comprise biomaterials derived from animal or plant tissues, such as proteins [77], polypeptides [78,79,80,81], polysaccharides [82], and nucleic acids [83]. The most commonly employed proteins in hydrogel include gelatin [84], collagen [85], and fibrin [86], while widely used polysaccharide polymers include alginate [87], chitosan [88], hyaluronic acid [89], and cellulose [90] (Figure 3b). Natural hydrogels are biocompatible and allow for minimal adverse reactions or immunogenic responses upon implantation or injection [91]. Moreover, natural hydrogels are also biodegradable and can be cleared from the body over time, reducing the risk of long-term complications [91]. Gelatin hydrogels have been widely used as drug-delivery systems for PDAC treatment. They undergo enzymatic degradation by proteolytic enzymes present in the body, allowing for their controlled biodegradation over time [92]. Currently, several natural polymers have been approved by the FDA for in vivo applications [93,94]. In addition, semi-synthetic hydrogels are composed of both natural and synthetic polymers, providing a combination of biocompatibility and controllability [95].

Generally, hydrogels are often porous networks formed through the chemical and/or physical crosslinking of polymers with a large amount of water [71]. The high water content of hydrogel is similar to that of native tissues, which could be beneficial for the adhesion and retention of hydrogel at the target site [70]. Meanwhile, the formulation, concentration, crosslinking type, and crosslinking degree of polymers can be modulated to generate various hydrogels with different physical and chemical properties [96]. Additionally, polymers can be crosslinked to form hydrogels of varying sizes, including macroscopic gels, microgels (0.5–500 μm), and nanogels (<200 nm). The diverse range of sizes endows hydrogels with distinct features and functionalities, consequently determining the optimal delivery route for cancer treatment [97]. Macroscopic gels with sizes in the scope of millimeters are often administrated locally for PADC through intratumoral injection, surgical implantation, peritumoral injection, and subcutaneous injection. For injection administration, most macroscopic gels were gelated in situ after injection. Meanwhile, microgels with a large surface area can be intraperitoneally or intratumorally injected to treat PDAC. In addition, nanogels can be intravenously injected to circulate in the body and permeate into tumor tissues [98] (Table 1).

Due to their distinct physical and chemical properties, hydrogels can deliver various therapeutics to treat diseases, including small-molecule drugs, proteins, nucleic acids, and cell preparations [97,99]. The high-water content and the porous networks of hydrogels endow them excellent capacities for efficiently encapsulating small- and high-molecular drugs. Drugs can be loaded into hydrogels through physical and chemical interactions. On the one hand, hydrophilic drugs could be physically entrapped in hydrogels by directly blending with the hydrogel polymers. While hydrophobic drugs are often loaded in hydrophilic micro/nanoparticles to improve their water solubility, and are therefore further entrapped within the hydrogels [100]. Moreover, hydrogels containing hydrophobic components (such as aliphatic chains and cyclodextrin) might also serve as physical binding sites for hydrophobic drugs [97]. Additionally, drugs could also be encapsulated through the electrostatic interactions between drugs and the polymers. Gaowa et al. reported that the EGFR2R-lytic peptide could form a complex hydrogel with gelatin through charge-based interactions [101]. On the other hand, drugs can also be immobilized within hydrogels through covalent interactions between drugs and polymers, such as amide bonds, thiol-ene bonds, ester bonds, and disulfide bonds [102]. As a particular drug-delivery system, hydrogel network can protect drugs from hydrolysis, inactivation, and enzymatic hydrolysis by impeding the rapid penetration of various enzymes, thus providing prolonged drug release [103,104]. In hydrogels, drug release typically occurs through multiple mechanisms, encompassing diffusion from the porous network [105], degradation of the network [106], hydrogel swelling [107], and deformation of the network [108]. For diffusion, the drug release rate of hydrogels is primarily influenced by the drug size/hydrogel mesh size ratio. Specifically, the hydrogel mesh size represents the dimension of the open space within the porous network. Correspondingly, drug size pertains to the physical dimensions of encapsulated drugs. A smaller drug size/mesh size ratio might result in rapid diffusion and release of drug from the porous network [97,109]. When the size of the drug approaches that of the mesh, the rate of drug release is considerably reduced [110]. When the drug size is larger than the pore size, the drug release would mainly depend on the degradation, deformation, and swelling of hydrogel [111]. During these processes, the hydrogel mesh size will be greatly increased, which controls the release rate of macromolecular drugs. Notably, except for the inherent properties of hydrogels, the degradation, deformation, and swelling of hydrogel can also be sensitive to various external stimuli, including pH [18], temperature [112], light [113], ultrasound [114], electric field [115], and the enzymes [116] and biomolecules [117] in body. Bilalis et al. developed pH- and enzyme-responsive polypeptide hydrogels which could release drugs at tumor sites through a solid-to-liquid transition [18]. Yan et al. reported a miRNA 21-responsitive hydrogel which would switch into a liquid state to release drugs when encountered with the overexpressed miRNA21 in TME [117]. Huang et al. demonstrated that the thermosensitive poly(N-isopropyl acrylamide) (PNIPAM) hydrogel could contracted to release GEM and H_2_S under the ultrasound stimuli [114]. Thus, introducing environment-responsive components into hydrogels could realize a controlled and sustained drug release to the target tumor sites (Table 1). 

### 3.1. Hydrogel-Based Small-Molecule Drug Therapy for PDAC

Hydrogels can encapsulate small-molecular drugs through physical blending, non-specific hydrophilic properties, or specific interactions with the drug. The design of hydrogel-based small-molecular drug therapy mainly focuses on drug combination strategies and the parameters which govern the release of the cargo [118,119]. Chemotherapeutics remain the most used small-molecule drugs in hydrogel-based delivery systems for PDAC treatment. The systemic administration of chemotherapeutics in clinical practice often leads to severe side effects [120]. Hence, employing hydrogels as a localized delivery system for chemotherapeutics might have the potential to reduce the adverse effects while maintaining therapeutic efficacy [121]. In the following section, we will discuss the strategies for small-molecule drug combination and drug release in hydrogel-based therapeutics for PDAC.

#### 3.1.1. Hydrogel as the Platform for Synergistic Therapy

The poor outcome of traditional therapy including chemotherapy and radiotherapy in advanced PDAC has provided opportunities for combined therapy, and hydrogels can exist as an appropriate platform to deliver drugs of different types to the tumor tissue [121]. In recent years, acoustic dynamic therapy (SDT) has been rapidly developed as an emerging noninvasive treatment for cancer. SDT eradicates tumor cells by generating ROS through the involvement of sonosensitizers and oxygen under ultrasonic radiation. This approach has garnered significant attention due to its ability to penetrate deeply into tissues [122]. However, the hypoxia TME of PDAC poses a challenge to the efficacy of SDT due to the rapid and substantial consumption of oxygen during therapy [123]. To address this issue, Huang et al. utilized microfluidic technology to fabricate a microcomposite hydrogel comprising an alginate shell and a perfluorocarbon (PFC) core. The resulting microgel featured a PFC-based core carrying oxygen and an alginate shell encapsulating GEM and indocyanine green (ICG). Following injecting into cancer-organoid-derived xenograft models, the PFC underwent a phase transition from liquid to gas upon low-density ultrasound stimulation in vitro, leading to the release of oxygen. Simultaneously, the acoustic-sensitive ICG generated a substantial amount of ROS under ultrasound stimulation, which penetrated the tumor tissue along with the GEM release from the alginate hydrogels. Consequently, this strategy for PDAC treatment could reverse the hypoxic microenvironment and induce apoptosis of tumor cells, and resulting in synergistic therapeutic effects compared to the GEM group (Figure 4) [124].
pharmaceutics-15-02421-t001_Table 1Table 1Hydrogel-based therapeutics for PDAC.HydrogelDrugHydrogel SizeDelivery RouteCharacteristicsAntitumor EffectRef.DNAAnti-miRNA21 antisense nucleic acid, GEMNanogelUnreportedA miRNA 21-responsive hydrogel which could simultaneously release drug and anti-miRNA. Inducing the apoptosis of tumor cells by targeting miRNA21.[117]OCMS, CMCSGEMMacroscopic hydrogelIntratumoral injectionAn injectable and thermosensitive hydrogel to sustainably release GEM.Inducing the apoptosis of tumor cells.[125]AlginateTumor cell lysate, GM-CSFMacroscopic hydrogelSurgical implantationA personalized hydrogel vaccine which sustainably released drug through the porous stereo structure. Recruiting DCs and enhancing the targeted antitumor immune response of CD8+ T cells.[126]AlginateGEM, ICGMicrogelIntratumoral injectionA core–shell microcapsule which can release oxygen and drug in presence of low intensity ultrasound. Hypoxic microenvironment reverse and apoptosis of PDAC cells activated by ROS.[124]ChitosanIRF5 mRNA, CCL5 siRNAMacroscopic hydrogelIntratumoral injectionAn in situ-injectable thermosensitive hydrogel with sustained RNA release.Inducing macrophage polarization and increasing the infiltration of CD8+ T cells into the TME, thus reshaping the immunosuppressive TME.[127]GelMAGEMMacroscopic hydrogelSurgical implantationAn adhesive microneedle patch that could efficiently penetrate the tumor tissue to release GEM.PDAC cell apoptosis.[128]Alginate, PLAGEMMacroscopic hydrogelSurgical implantationA hydrogel patch with reduced swelling ratio exhibiting prolonged drug release. PDAC cell apoptosis[75]PDLLA-PEG-PDLLAGEM, DPP-BTzMacroscopic hydrogelIntratumoral injectionThermosensitive liposomal hydrogels with NIR-II light-triggered drug release. PDAC cell apoptosis[129]PNIPAM, Alginate, PVAGEM, H_2_SMicrogelIntratumoral injectionUltrasound responsive microbubble hydrogel, which contracted under the increasing temperature resulted from ultrasound, thus releasing GEM and H_2_S.Contributing to PDAC cell apoptosis and inhibiting PDAC cell proliferation. [114]HANeoantigen peptideMacroscopic hydrogelSurgical implantationA hydrogel vaccine with sustained adjuvant release.Enhanced T-cell activation in the draining lymph node and expansion of neoantigen-specific T cells in the spleen.[130]Alginate GEM or DOXMacroscopic hydrogelSurgical implantationCoaxial hydrogel fibers exhibiting a slower release profile due to the core–shell structure for controlled release and diffusion barrier.Inhibiting the growth of PDAC cells.[131]PNIPAM, CS, PEG, GNRUnreportedMacroscopic hydrogelIntratumoral injectionA thermal-sensitive hydrogel which shrunk with the increased temperature induced by an 808 nm laser. Inducing tumor internal stresses, hypoxia, and apoptosis.[19]PDLLAPEG-PDLLAGEM, cisplatinMacroscopic hydrogelIntratumoral injectionA thermal-sensitive hydrogel gelated in situ at physiological temperature, exhibiting delayed drug release from the micelle networks.Inducing PDAC cell apoptosis and inhibiting proliferation.[132]TerpolypeptideGEMMacroscopic hydrogelPeritumoral injectionA self-healing hydrogel that can deliver drugs sustainably due to its pH- and enzyme-responsive nature. PDAC cell apoptosis.[18]PCLA-PEG-PCLA)GEMMacroscopic hydrogelSubcutaneous injectionA thermal-sensitive nano-biohybrid hydrogel with sustained drug release. PDAC cell apoptosis.[133]PoloxamerPTXMacroscopic hydrogelIntratumoral injectionA thermosensitive hydrogel with paclitaxel liposome showed a slower release than liposome.Unreported[134]PLGA-bPEG-b-PLGADOXMacroscopic hydrogelIntratumoral injectionA thermosensitive hydrogel with micelle networks.Unreported[135]PEG, HSATRIALMacroscopic hydrogelIntratumoral injectionA PEG-modified albumin hydrogel, gelated in situ.PDAC cell apoptosis.[136]GelatinEGFR-lyricNanogelIntravenous injectionHydrogel nanoparticles formed by electrostatic interaction exhibiting a longer circulation time in vivo.Unreported[101]HATRIALMacroscopic hydrogelIntratumoral injectionPEG-TRAIL HA hydrogels with stability and controlled drug release.PDAC cell apoptosis.[137]PVADOX, mitoxantrone, irinotecanMicrogelIntraperitoneal injectionDrug eluting hydrogel beads.PDAC cell apoptosis.[138]PEG-PCL-PEGLPS, FGFMacroscopic hydrogelSubcutaneous injectionsA hydrogel vaccine with adjuvant release.Enhancing both cellular and humoral immune response against PDAC.[139]


In addition to SDT, another novel approach with antitumor potential is photodynamic therapy (PTT), which utilized the photothermal effect induced by near-infrared light [140]. To enhance the effectiveness of PTT, thermosensitive hydrogels can be used to achieve thermally responsive drug release. Yingjie, Kong et al. developed an injectable thermosensitive hydrogel designed to deliver nanoparticles of GEM and PTT sensor DPP-BTz to the TME. In this study, dipalmitoyl phosphatidylcholine (DPPC) was employed to synthesize thermosensitive liposomes loaded with GEM and a photosensitizer DPP-BTz. Additionally, poly(D,L-lactide)-polyethylene glycol-poly(D,L-lactide) (PLEL) was utilized to prepare thermosensitive hydrogels encapsulating the aforementioned liposomes. Upon injection into the tumor tissue of a murine PDAC model, the hydrogel solution underwent a gelation process triggered by the body temperature of the mouse. Subsequently, near-infrared light (1064 nm) was applied in vitro, initiating the photothermal effect mediated by the photosensitizer. This effect led to a local temperature increase, which disrupted the structure of the thermosensitive liposomes and released GEM into the hydrogel. Consequently, GEM diffused into the TME, effectively inducing apoptosis of PDAC cells [129].

The delivery of different drugs in combination therapy can be realized by the hydrogel systems. Huang, D. et al. developed an ultrasound-responsive microbubble hydrogel for the simultaneous delivery of hydrogen sulfide (H_2_S) gas and GEM into PDAC TME. Using microfluidic electrospray technology, they fabricated a hydrogel precursor composed of an inner core containing H_2_S gas and an outer shell of alginate and PNIPAM loaded with GEM. The hydrogel precursor was crosslinked using calcium ions and ultraviolet radiation. Upon injection of the hydrogels into the PDAC tissue of mice, in vitro ultrasonic stimulation caused expansion and oscillation of the H_2_S gas within the inner layer of the hydrogel, resulting in an increase in the surrounding temperature. Simultaneously, the outer layer of the hydrogel contracted, leading to the release of H_2_S and water-soluble GEM into the neighboring tumor tissue, exerting their antitumor effect [114]. Moreover, Shi, Kun et al. developed an injectable hydrogel to enhance the potential of chemotherapy combination therapy for PDAC. They utilized an amphiphilic triblock copolymer PLEL as the hydrogel prepolymer, which is capable of self-assembling into core–shell-like micelles in water at room temperature. When exposed to body temperature, the micelles formed a micellar network through spontaneous crosslinking. This hydrogel exhibited delayed GEM and cisplatin release from the micelle networks, thereby inducing apoptosis and inhibiting the proliferation of PDAC cells more significantly compared to the GEM alone [132].

#### 3.1.2. Encapsulating Drugs in Microneedles (MNs) to Achieve Sustained Release

MNs possess the remarkable ability to seamlessly penetrate tissues, offering the potential for targeted, long-lasting, and widespread drug release. Fu, X et al. utilized gelatin methacryloyl (GelMA) to fabricate a hydrogel MN adhesive patch, which could enhance the release of GEM into the PDAC TME. Inspired by the tentacles of an octopus, the study incorporated grooves on the base of the MNs to mimic the suction cups found in biological organisms, thus improving the tissue adhesiveness of the MN patches. The MN hydrogels were created through mold-based ultraviolet irradiation. Following implantation into the PDAC tissue, the MNs seamlessly penetrated deep within the tumor. Over a period of 7 days, the MNs gradually degraded. This allowed for the gradual release of the encapsulated GEM into the tumor tissue, resulting in a more extensive and effective eradication of neighboring PDAC cells compared to GEM administered alone. Furthermore, the drug-release kinetics could be modulated by adjusting the concentration of GelMA [128].

#### 3.1.3. Design of TME-Responsive Hydrogel Degradation to Achieve Sustained Release

Peptide hydrogels, serving as drug-delivery systems, possess the ability to encapsulate drugs and release them by the modulating their structure and composition. Additionally, they exhibit excellent biocompatibility and can be easily synthesized [141]. Moreover, peptide hydrogels display various responsiveness, allowing for precise regulation of drug release in response to external environmental factors [142]. Bilalis et al. developed a hydrogel for PDAC treatment by combining a unique pentablock terpolypeptide (PLys-b-(PHIS-co-PBLG)-PLys-b-(PHIS-co-PBLG)-b-PLys) with GEM in a syringe. The hydrogel was designed to specifically deliver the drug to cancerous tissue while minimizing the harm inflicted upon healthy tissue. The distinctive macromolecular architecture of the polypeptide facilitated spontaneous gelation of the solution upon injection into the tumor tissue without the need for external stimulation. Simultaneously, the polypeptide hydrogel underwent degradation in response to the acidic pH and protease levels within the TME of PDAC, thereby resulting in the gradual release of GEM and effectively inducing apoptosis of PDAC cells (Figure 5) [18].

#### 3.1.4. Incorporating the Drugs into the Polymer Network of Hydrogel to Release Sustainably

Drugs can also be integrated into the polymer network by means of dopamine functionalization to achieve controlled release. Xu, L et al. designed a hydrogel by incorporating derivatives of oxidized-carboxymethylcellulose (OCMC) and carboxymethyl chitosan (CMCS), which are both polysaccharides functionalized with dopamine. In vivo experiments showed that this hydrogel was capable of retaining GEM for 7 days, facilitated by the interaction mediated by the catechol groups of dopamine. Additionally, the presence of GEM also increased the swelling ratio of the drug-loaded hydrogel, potentially due to its disruption to the hydrogen bonds between the polymers [125,143,144]. Sepehr, Talebian et al. capitalized on the affinity between dopamine and drugs to develop an alginate hydrogel which effectively released chemotherapy drugs, GEM or doxorubicin (DOX), for PDAC treatment. To prevent burst release and enhance controlled diffusion, an outer layer of double-crosslinked alginate hydrogel was applied as a barrier to the hydrogel core. Methacrylic anhydride was used to modify the alginate, allowing for secondary crosslinking under ultraviolet irradiation after calcium-mediated crosslinking. The resulting bilayer hydrogel exhibited superior mechanical properties compared to the monolayer hydrogel, and displayed a slower drug release rate and improved therapy outcome compared to GEM alone [131]. Subsequent studies by the same team optimized the mechanical properties and swelling rate of the bilayer alginate hydrogel. Utilizing 3D-printing technology, they prepared a four-layer hydrogel patch by incorporating calcium carbonate (CaCO_3_) and polylactic acid (PLA) as wrapping layers. This four-layer hydrogel patch demonstrated continuous drug release for 7 days in vitro, surpassing the performance of the double-layer hydrogel [75].

#### 3.1.5. Encapsulating the Drugs into Hydrophobic Nanoparticles to Achieve Sustained Release

The delivery of hydrophilic small molecule drugs by hydrogels can result in rapid and uncontrolled release. To address this issue, the drugs can be enclosed within hydrophobic nanoparticles to entrap them within the hydrophilic hydrogel network, thereby slowing down the release rate. To achieve controlled release of GEM, Phan et al. developed an injectable, thermosensitive nano-biohybrid hydrogel. GEM was incorporated into the interlayer gallery and surface of montmorillonite (MMT) nanoparticles, forming MMT-GEM complexes. These complexes were then dispersed into biodegradable and thermosensitive solutions composed of poly (ε-caprolactone-co-lactide)-b-poly (ε-caprolactone-co-lactide) (PCLA-PEG-PCLA), which could undergo gelation at the physiological temperature. Compared to the original hydrogel, the adsorption of GEM in MMT nanoparticles significantly slowed down the release rate of GEM, leading to a significant reduction in initial release. Furthermore, the nano-biohybrid hydrogel network exhibited a reduced mesh size compared to the original hydrogel, leading to a sustained release of GEM and long-term inhibition of PDAC growth [133]. Mao et al. also utilized a thermosensitive Poloxamer hydrogel to encapsulate paclitaxel liposomes (PTX-lip) for the treatment of PDAC in mice. Following injection into the PDAC tissue, the PTX-lip loading hydrogel group demonstrated enhanced drug retention and longer survival of mice with PDAC, compared to the PTX liposome group [134].

### 3.2. Hydrogel-Based Nucleic Acid Therapy for PDAC

DNA, with its two- or three-dimensional structures, is capable of constituting a highly organized network through the crosslinking of complementary DNA molecules. When in contact with water, these DNA-based hydrogels exhibit impressive swelling and expansion properties. These hydrogels can encapsulate various nucleic acid molecules, including siRNA, miRNA, and drugs that can bind to DNA [145]. The hydrogels possess excellent solubility, biocompatibility, functionality, and responsiveness. Yan, J et al. devised a DNA hydrogel to encapsule GEM nanoparticles and an antisense oligonucleotide which specifically binds to microRNA 21. They grafted two acrylic-stone-modified DNA sequences onto linear polyacrylic acid (PAA) through radical reactions to generate the precursors for the hydrogel. GEM was loaded into mesoporous silica nanoparticles (MSN), and the aforementioned hydrogel precursors were attached to the surface of 3-(trimethoxysilyl)propyl methacrylate (TMSPMA)-modified MSN via free radical reaction. Then, an antisense nucleic acid, anti-miR-21, which is partially complementary to the two DNA sequences, was introduced as a crosslinking agent to promote the formation of nanoscale hydrogels. These hydrogels demonstrated responsiveness to the overexpressed miRNA-21 in the TME and could return to liquid state upon encountering miRNA-21, thereby facilitating targeted releasing GEM. Meanwhile, the anti-miRNA 21 could also silence miRNA 21, resulting in a synergistic therapeutic effect when combined with GEM [117].

Immunomodulatory factor 5 (IRF5) has been reported to promote the polarization of M2 TAMs into M1 phenotype in the TME. Conversely, CCL5 has been implicated in tumor progression through its involvement in TAM recruitment. Gao, C et al. exploited an injectable thermosensitive chitosan hydrogel to deliver IRF5 mRNA and CCL5 siRNA directly into PDAC TME. They employed protamine, a natural cationic protein, to form complexes with negatively charged RNA molecules, protecting mRNA from degradation by serum RNases. To enhance delivery efficiency, the protamine/RNA complex was further encapsulated with liposomes composed of 1,2-Dioleoyl-3-trimethylammonium-propane (DOTAP) and cholesterol. This RNA-loaded liposome system was then combined with chitosan to create the hydrogel precursor. Upon injection into mice, the hydrogel precursor was crosslinked at physiological temperature, forming a gel-like substance. Over time, the hydrogel gradually degraded within the body, leading to the release of the nanoparticles into the TME, thus facilitating RNA delivery to tumor cells and exerting a therapeutic role. Remarkably, within 16 days of implantation, more than 90% of liposomes were released from the hydrogel. This innovative hydrogel system effectively induced the polarization of M2 macrophages to the M1 phenotype and increased the infiltration of CD8+ T cells into the TME. Consequently, this approach holds great promise for preventing the recurrence and metastasis of PDAC following surgical intervention (Figure 6) [127].

### 3.3. Hydrogel-Based Protein Therapy for PDAC

When using hydrogels to deliver protein drugs, it should be noted that the functional groups of protein might interact with the polymer of hydrogel network and the crosslinking condition needs to be moderate so that the encapsulated proteins would not denature. Yi, Lu et al. fabricated a personalized hydrogel vaccine to prevent the recurrence of PDAC tumors after surgery. The authors mixed the lysate of PDAC cells obtained from the surgical resection of mouse PDAC samples with alginate and GM-CSF. The amino group in the lysate were crosslinked with the carboxyl group of alginates through amide bonds under the catalysis of 1-(3-Dimethylaminopropyl)-3-ethylcarbodiimide hydrochloride (EDC) and N-heterocyclic carbene (NHC) to obtain an immunogenic hydrogel. Transplantation of the hydrogel into the surgical area of mice could increase the local DC infiltration, thereby effectively preventing tumor recurrence in mice [126]. 

Delitto, D et al. synthesized a hyaluronic acid (HA) hydrogel to deliver the neoantigen peptide derived from the murine PDAC cell line Panc02. In the previous study, the authors performed whole exome sequencing of Panc02 cells and found 12 mutant epitopes which might bind to MHC I. Subsequently, they synthesized the peptide containing these 12 epitopes in vitro and utilized HA hydrogel to encapsule this neoantigen. After the surgery, this hydrogel was implanted into the tumor region. Consequently, a transient influx of MDSCs, a prolonged neutrophil influx, and a near-complete loss of cytotoxic T cells were demonstrated. Application of this gel was associated with enhanced T cell activation in the draining lymph node and expansion of neoantigen-specific T cells in the spleen. Finally, this hydrogel could effectively prevent the recurrence of the PDAC after incomplete surgery [130]. 

Tumor-necrosis-factor-associated apoptosis-inducing ligand (TRAIL) mediates cell apoptosis in a variety of tumor cells by specifically binding to death receptors that are highly expressed on the surface of tumor cells. TRAIL is promising for PDAC treatment, but the proteins are easily denatured and inactive by physical or chemical stimuli. Therefore, Hyeong, Byeon et al. used PEG to modify the TRAIL protein (PEG-TRAIL) and found that its antitumor effects and pharmacokinetics were better than TRAIL protein alone. The later research of this group used HA and PEG to develop hydrogels for delivering PEG-TRAIL protein to the mouse PDAC TME, which could significantly induce apoptosis of PDAC cells and inhibit tumor growth [137]. In the follow-up study, the team also used the combination of human albumin and PEG to obtain an injectable hydrogel precursor; the precursor solution spontaneously formed gel within 60 s after injection into the murine PDAC tissue [102].

Arong, Gaowa et al. synthesized an EGFR-lytic peptide for PDAC treatment based on the chemical coupling of lysing peptides and peptides targeting epidermal growth factor receptors (EGFRs) overexpressed on the surface of tumor cells. This peptide exhibited good efficacy in a variety of solid tumors. However, in vivo experiments showed that the peptide had a short blood residence time and multiple administrations were required to achieve an effective plasma concentration for exerting the curative effect. To address this issue, the authors prepared gelatin nanoparticles with a high permeability and a resident effect (EPR) to deliver EGFR-lytic peptides. Through electrostatic action, negatively charged gelatin formed nanoparticles with positively charged EGFR lysis peptide, which significantly inhibited the growth of PDAC in mice compared with EGFR lysis peptide after intravenous injection [101].

Lipopolysaccharides (LPS) derived from the cell wall of Gram-negative bacteria can be recognized by APCs, such as macrophages, DCs, and murine B lymphocytes, thus activating innate immunity, which are a commonly used immune adjuvant. Basic fibroblast growth factor (bFGF) is one of the most important antitumor angiogenic factors. Huashan et al. intercepted a segment (tbFGF) from the functional domain of bFGF, which could prevent the proliferation of endothelial cells in vitro, but not tumor cells. They further devised an injectable and thermosensitive poly (ethylene glycol)poly(N-caprolactone)-poly(ethylene glycol) (PEG-PCL-PEG, PECE) hydrogel to deliver LPS and tbFGF to enhance the immunogenicity of tumor-associated antigens, thereby enhancing the immune response and improving the therapeutic effect of cancer. The results showed that the delivery system could promote antibody-mediated and CTL-mediated immune response, increase the secretion of γ-interferon and IL-4, and inhibit tumor growth and metastasis in vivo [139].

## 4. The Potential Drug Candidates for Hydrogel-Based Immunotherapeutic Options for PDAC Treatment

Several clinical trials are currently underway to investigate potential therapeutic approaches for PDAC through targeting the immunosuppressive TME. Notably, a combination of immunotherapy and chemotherapy has yielded promising results. An emerging approach involves the utilization of hydrogels as delivery systems, offering an efficient and safe means for administrating therapeutic agents. In the subsequent section, we will discuss potential drug candidates for hydrogel-based therapy targeting immunosuppressive stroma cells in the TME of PDAC (Table 2).

### 4.1. MDSCs Directed Therapy

The process involving the recruitment of MDSCs from the bone marrow to the TME shares similarities with TAMs. As such, this section will primarily focus on targeting strategies which are specific to MDSCs and discuss the therapeutic approaches which are available to inhibit their recruitment. For instance, among mice with GEM-resistant PDAC, it was identified that tumor cells secreted GM-CSF, leading to the production of MDSCs within the TME. This, in turn, hampered the function of CD8+ T cells. However, by administrating an anti-GM-CSF monoclonal antibody alongside GEM, it is possible to counteract the upregulation of MDSCs and restore the sensitivity of PDAC cells to GEM [14]. Hence, neutralizing the GM-CSF effect can effectively circumvent therapy resistance. Chemokines known as CXCLs, secreted by tumor cells during tumorigenesis, facilitate the attraction of neutrophils and PMN-MDSCs into the TME via the CXCLs-CXCR signaling axis. In murine models of PDAC, genetic knockout or pharmacological inhibition of CXCR2 effectively reduced tumor-associated neutrophilia (TANs) and PMN-MDSCs within the TME. This reduction in myeloid cell infiltration fostered increased T-cell infiltration and concurrently impeded tumor metastasis. Notably, the combination of CXCR2 inhibitors with immune checkpoint inhibitors has shown significant prolongation of the survival of PDAC-bearing mice [146]. Timothy also reported that, in the orthotopic murine PDAC model, the CXCR2 blockade inhibited the infiltration of PMN-MDSCs and enhanced the efficacy of chemotherapy [149]. Radiotherapy has been associated with an increased number of MDSCs in tumor tissues of PDAC patients, which corresponds to resistance to radiotherapy efficacy. Researchers have identified the key role of STAT3 in this phenomenon and successfully improved radiotherapy efficacy in murine PDAC models by inhibiting STAT3 using antisense nucleotides [51]. In a murine PDAC model, Dmitry et al. first discovered that the TRAIL receptor (TRAIL-R) specifically mediates MDSC apoptosis within the TME, making it an ideal target for MDSC-centered therapies. Subsequently, they developed the TRAIL-R agonist DS-8273a for the treatment of PDAC patients. They observed a reduction in MDSC level in the patients’ peripheral blood, while the number of neutrophils, monocytes, and lymphocytes remained unaffected. In addition, DS-8273a treatment significantly reduced MDSC infiltration in the tumor tissues of PDAC patients [147].

### 4.2. TAMs-Directed Therapy

Therapeutic strategies targeting TAMs in PDAC have shown potential. These strategies primarily involve inhibiting TAM infiltration in the TME and reprogramming TAMs towards antitumor phenotypes. A critical factor in PDAC development is the release of chemokines like CCL2 and CSF1, which recruit monocytes from the peripheral blood and differentiate them into TAMs within the TME. Therefore, they block the CCL2/CSF1-CSF1R axis, which theoretically reduces TAM infiltration. An investigational single-arm clinical study assessed the efficacy of the CCL2 inhibitor CCX872-B in combination with FOLFIRINOX chemotherapy for treating locally advanced and metastatic PDAC. The results demonstrated an 18-month overall survival (OS) rate of 29% in the combination group, surpassing the 18-month OS rate (18.6%) reported in the literature for the FOLFIRINOX treatment group [148]. However, further clinical trials are necessary to validate the effectiveness of CCX872-B in PDAC. Furthermore, studies have highlighted the immunosuppressive nature of TAMs in PDAC. Inhibition of CSF1R signaling has been found to decrease TAM numbers and enhance their antigen-presentation ability, ultimately promoting antitumor T-cell immunity [150]. Notably, CD47, expressed on the surface of PDAC cells, mediates a “do not eat me” signal between tumor cells and macrophages [160]. Blocking this signal with a CD47 monoclonal antibody has been shown to significantly enhance TAMs’ phagocytosis, leading to increased CD8+ T-cell populations and activated subsets in the PDAC TME [151,161,162,163]. A deeper investigation into the mechanisms of CD47-mediated macrophage avoidance among tumor cells unveiled that the use of CpG oligodeoxynucleotides, a Toll-like receptor 9 agonist, induced metabolic changes in TAMs. These changes reversed the CD47-mediated “do not eat me” signal and facilitated the phagocytosis of tumor cells [152].

### 4.3. CAFs-Directed Therapy

The exploration of PDAC biology in recent years has intensified the investigation into the role of CAFs in the pathogenesis and progress of PDAC, paving the way for the development of targeted therapies against CAFs. Given the protumor functions attributed to CAFs, strategies aimed at the elimination of these cells have emerged as potential therapeutic interventions. In the context of PDAC, α-SMA and FAP have been identified as key markers for CAFs within the TME. Subsequently, the targeting of small-molecule agents, such as IPI926 and Halofuginone, has been developed. This strategy has demonstrated a significant reduction in the infiltration of CAFs in murine PDAC models, resulting in the improved survival of tumor-bearing mice. Despite these promising preclinical results, the translation of these therapies into clinical practice has faced challenges due to evident side effects, leading to discontinuation of the treatment in clinical trials [153,154,164]. Moreover, it is important to recognize that α-SMA and FAP are also highly expressed in normal tissues, including skeletal muscle and bone marrow, casting concerns over potential off-target effects when employing CAF-depletion strategies which target both markers simultaneously, as it may lead to severe adverse consequences, such as cachexia and anemia [165]. To address this limitation, hydrogel-based delivery systems can emerge as an innovative approach to minimize damage to normal tissues. Through localized and precise administration of therapeutic agents, hydrogels offer a potential means to compensate for the drawbacks of CAF elimination strategies, enhancing the therapeutic efficacy while minimizing the systemic toxicity associated with the treatment. 

Targeting CAFs by inhibiting the transformation of PSCs into CAFs represents an additional therapeutic approach for PDAC. PSCs are considered a source of CAFs within the TME of PDAC, and several signaling pathways, including JAK/STAT3 and mTOR, play significant roles in phenotypic transitions. Consequently, targeted drugs that modulate these pathways have been subject to investigation. For instance, Yu Shi et al. demonstrated that the paracrine regulator leukocyte inhibitor (LIF), produced by CAFs, plays a critical role in activating the STAT3 signaling pathway in PDAC. Genetically knocking out the LIF receptor in pancreatic epithelial cells resulted in a significant extension of survival in pancreatic-oncogene-bearing mice. Complementary studies revealed that the use of a LIF mAb in combination with chemotherapy considerably reduced the pancreatic oncogene-induced tumor growth in mice compared to chemotherapy alone [155]. In a similar vein, Siham et al. identified the somatostatin analog SOM230 as an inhibitor of protein synthesis and secretion in CAFs, acting through the mTOR/4E-BP1 signaling pathway. This inhibition led to the effective suppression of PDAC metastasis in murine models [156]. Furthermore, vitamin D has shown promise in reversing PSC activation, maintaining them in a quiescent state and delaying PDAC progression in murine models. Mara H et al. discovered a high expression of vitamin D receptor (VDR) expression in the matrix of human PDAC. Treatment with the vitamin D analogue calcipotriene induced a remodeling of the PDAC matrix. Combined administration of calcipotriene and the antitumor drug GEM resulted in significantly increased gemcitabine concentration, reduced tumor volume, and enhanced survival rate by 57% compared to gemcitabine alone [157]. Nintedanib, a common antifibrotic drug, exerts its effects by blocking the platelet-derived growth factor receptor β (PDGFRβ) signaling pathway, leading to reduced CAF activation and growth. In combination with other targeted therapies such as MEK inhibitors or CAR-NK, nintedanib demonstrated effectiveness in inhibiting PDAC growth in murine models [158,166].

Researchers are also working to approach the treatment of PDAC by focusing on the manipulation of the extracellular matrix structure generated by CAFs. To this end, pegylated recombinant human hyaluronidase PH20 (PEGPH20) has been employed to degrade HA within a GEMM model of PDAC. Subsequently, the impact on tumor blood vessels, including perfusion, permeability, and drug delivery capacity, was evaluated. The study revealed that PEGPH20 effectively and persistently degraded HA in tumors, resulting in the dilation of PDAC blood vessels and subsequently increasing the concentration of chemotherapeutic agents within the tumor tissue [167]. Moreover, PEGPH20 was shown to increase the intracellular gap on the PDAC vascular endothelial cells, specifically enhancing the macromolecular permeability of tumor blood vessels. Combined with GEM, PEGPH20 exhibited superior inhibitory effects on PDAC growth and prolonged the survival of tumor-bearing mice, thus highlighting the clinical potential of PEGPH20 [168]. Inspired by promising results from a phase I study, the phase III HALO-109-301 study aimed to investigate the efficacy of PEGPH20 in combination with chemotherapy for previously untreated stage IV PDAC patients. Unfortunately, the trial was prematurely terminated because the experimental group did not meet the primary endpoint. Nevertheless, the experimental group did demonstrate higher response rates, although no significant improvements were observed in terms of response duration and OS [159]. These findings suggest that exploring the combination of PEGPH20 with immunotherapeutic agents targeting immune cells may yield fruitful avenues for further investigation.

## 5. Discussion and Prospectives

Although the progress in PDAC treatment has been limited, preclinical models have demonstrated promising outcomes with novel therapies that target various components within the immunosuppressive TME of PDAC. However, clinical trials have generally failed to meet expectations, potentially due to the side effects of systemic administration [169]. This predicament has created an opportunity for the utilization of delivery systems in PDAC treatment. Hydrogels have emerged as a specific drug-delivery system capable of encapsulating both small-molecule and large-molecule drugs in a straightforward manner. The advent of responsive hydrogels has enabled the development of injectable hydrogels and hydrogels sustainably releasing pharmaceutics [97,99,103,104]. Moreover, hydrogels offer improved safety and biocompatibility, with reduced toxic side effects [91,92]. These advantages position hydrogels as potential delivery systems for PDAC treatment.

In the realm of PDAC, hydrogel-based therapies have emerged with promising potential across multiple domains. This article provides a comprehensive introduction of hydrogel-based treatments for PDAC and observes that the majority of these interventions are centered on chemotherapeutic drugs. By incorporating chemotherapeutic drugs within the network of hydrogels, drug retention within the tumor can be enhanced, leading to the improvement of local drug concentration while simultaneously mitigating systemic adverse effects. In addition to their role in chemotherapy, hydrogels can also be customized to incorporate biological molecules or immune modulators, furnishing an avenue to stimulate immune cell infiltration and reprogram the immunosuppressive TME of PDAC.

Hydrogel-based therapies offer promising strategies for the treatment of PDAC, yet several challenges remain for future advancements in this field. Firstly, it is crucial to develop more effective strategies to remodel the immunosuppressive TME of PDAC. Recent research has highlighted the close relationship between stroma cells and the resistance of PDAC to chemotherapy, radiotherapy, and other drugs, consequently sparking growing interest in therapeutic approaches targeting the immunosuppressive TME of PDAC. Small molecule inhibitors and monoclonal antibodies have emerged as potential options for PDAC treatment, and their combination with hydrogels holds promise for immunotherapy strategies which can effectively reshape the TME. Additionally, hydrogel-based tumor vaccines and cell therapies, such as CAR-T, may provide valuable strategies to specifically target and reverse the immunosuppressive TME. Furthermore, while hydrogel-based chemotherapies have predominantly focused on GEM, the potential application of other drugs, such as DOX, in hydrogel-based PDAC treatment should be explored. Secondly, the effectiveness of a single hydrogel-based therapy may not meet requirements. Despite the emergence of various therapies which have demonstrated efficacy in preclinical models or clinical trials, their utilization in hydrogel-based PDAC treatment remains relatively limited. Future progress should involve more combined hydrogel-based therapies, such as the integration of chemotherapy and immunotherapy targeting the immunosuppressive TME. Thirdly, the development of “smart” hydrogel-based delivery system capable of targeted and controlled release of drug is imperative for effective PDAC treatment. Presently, most hydrogels release drugs through physical diffusion. To enhance the therapeutic outcome, hydrogels should be designed to be more responsive to the PDAC TME or controlled external stimulus. For example, Wei, Hai et al. fabricated a DOX-loaded nanosystem based on nucleic acid which formed a hydrogel through the cross-linking of polyacrylamide chains employing nucleic acid hairpins. This hydrogel selectively released the drug in the presence of ATP [170]. In addition, the development of biomaterials suitable for personalized PDAC therapy, such as autologous biomaterials, is essential. Notably, implanted autologous blood clot scaffold have demonstrated the ability to induce robust anticancer immune response as an enhanced cancer vaccination [171]. Consequently, the potential utilization of autologous plasma hydrogels could be explored as a novel strategy for personized immunotherapy in PDAC. Finally, to pave the way for the clinical application of hydrogel-based PDAC therapies, multidisciplinary collaborations are imperative to ensure their safety and effectiveness as viable alternatives to the traditional treatments.

Overall, the application of hydrogels in the context of PDAC presents intriguing prospects for targeted and personalized therapeutic interventions. Leveraging their distinctive characteristics, hydrogels can potentially overcome the issues related to drug delivery and resistance, thereby enhancing treatment outcomes for individuals diagnosed with PDAC.

## Figures and Tables

**Figure 1 pharmaceutics-15-02421-f001:**
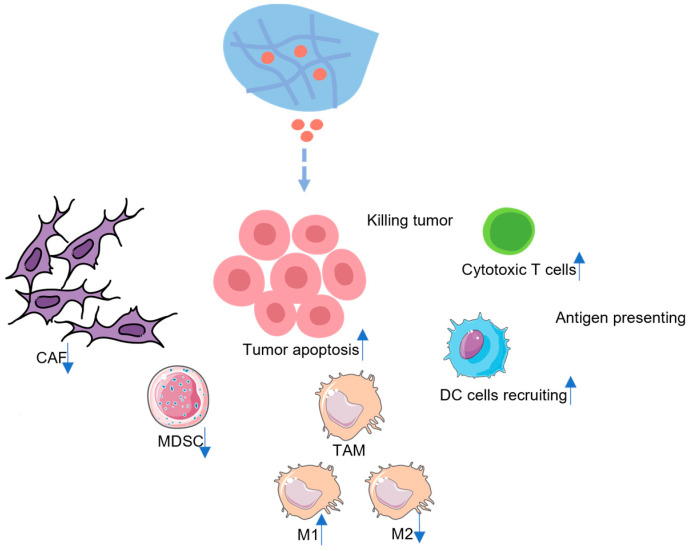
The mechanism of hydrogel-based therapeutics for PDAC treatment.

**Figure 2 pharmaceutics-15-02421-f002:**
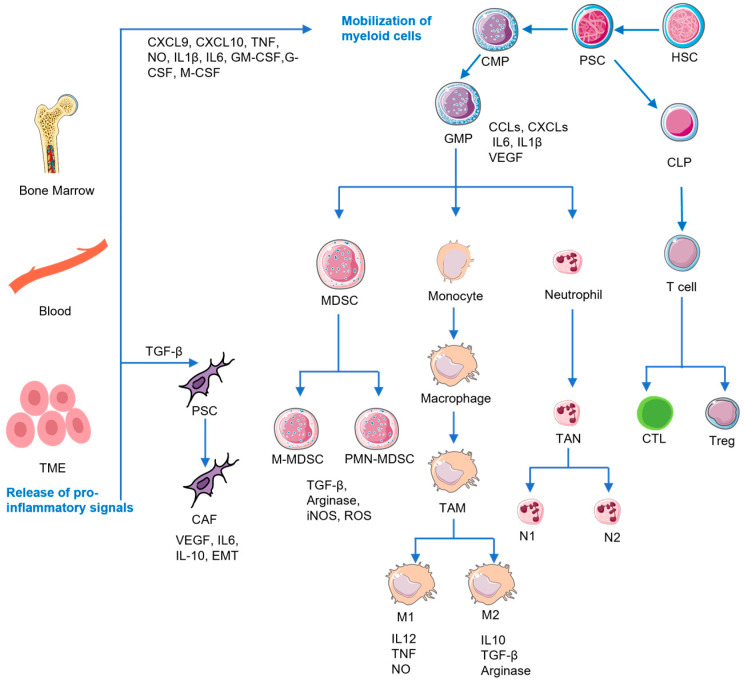
The formation of the immunosuppressive TME of PDAC.

**Figure 3 pharmaceutics-15-02421-f003:**
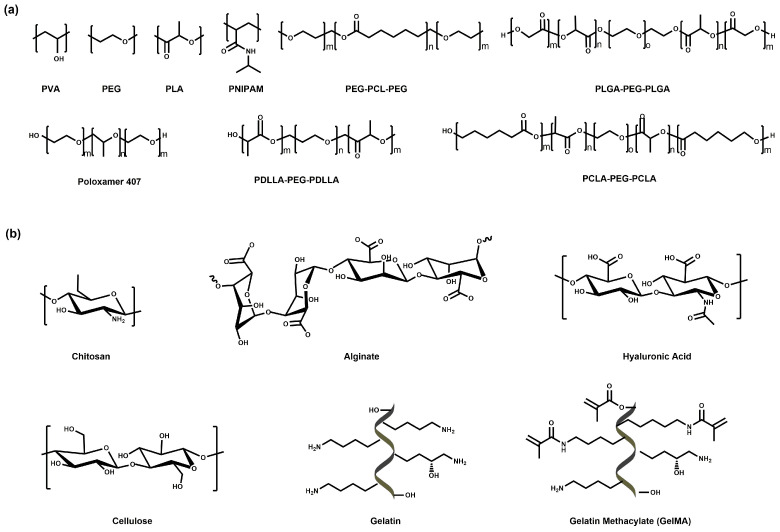
The chemical structure of typical synthetic (**a**) and natural (**b**) polymers for hydrogel-based delivery systems in PDAC.

**Figure 4 pharmaceutics-15-02421-f004:**
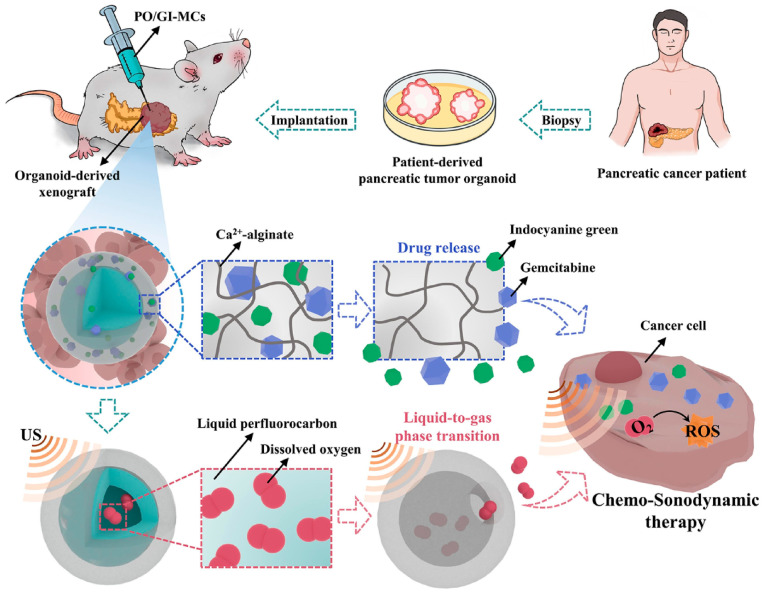
A schematic illustration is presented to demonstrate the application of PFC-alginate hydrogel delivering GEM and ICG on a xenograft model derived from patient-derived organoids of PDAC. The hydrogel composite was utilized to deliver and release chemotherapeutics and sonosensitizers at the tumor site via in situ injection. Upon exposure to low-intensity ultrasound, the liquid-phase PFC underwent a phase transition into a gas, leading to the release of dissolved O_2_, thereby enhancing the efficacy of sonodynamic therapy. Copyright from ELSEVIER 2023.

**Figure 5 pharmaceutics-15-02421-f005:**
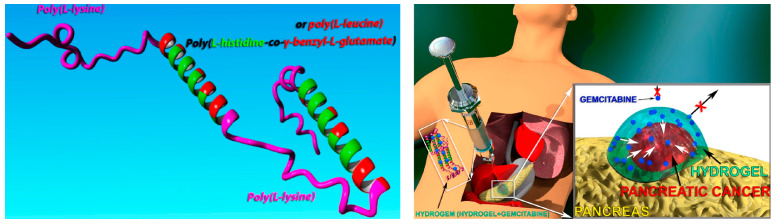
The structure of pentablock terpolypeptide and the illustration of the hydrogel with encapsulated drug implanted in least invasive way. The hydrogel could melt only in vicinity of cancerous tissue due to the lower pH, leading to targeted and directional drug delivery. Copyright from ACS Publications 2023.

**Figure 6 pharmaceutics-15-02421-f006:**
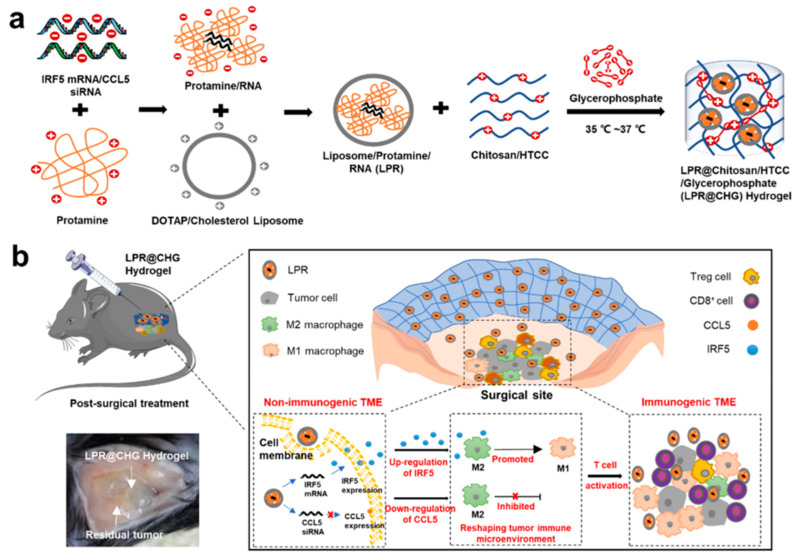
The schematic illustration depicts the process of formulating the chitosan-based hydrogel (**a**) and highlights the underlying mechanism of reshaping the tumor microenvironment for the treatment of pancreatic cancer after surgery (**b**).

**Table 2 pharmaceutics-15-02421-t002:** Potential drugs targeting the immunosuppressive TME for PDAC.

Drug	Therapeutic Target	Preclinical Model/Clinical Trial	Publish Year	Ref.
anti-GM-CSF monoclonal antibody	PMN-MDSCs	Subcutaneous transplantation tumor in mice	2023	[14]
CXCR2 inhibitors	PMN-MDSCs	GEMM	2016	[146]
STAT3 antisense nucleotides	PMN-MDSCs	Subcutaneous transplantation tumor in mice	2021	[51]
TRAIL-R agonist DS-8273a	PMN-MDSCs	Phase I trial	2017	[147]
CCL2 inhibitor CCX872-B	TAMs	Phase Ib trial	2017	[148]
CCR2 inhibitors	TAMs	Orthotopic transplantation tumor in mice	2018	[149]
CSF1R blockade	TAMs	GEMM	2014	[150]
anti-CD47 monoclonal antibody	TAMs	Hepatic micro-metastatic tumor model in mice	2018	[151]
CpG oligodeoxynucleotides	TAMs	Orthotopic transplantation tumor in mice	2019	[152]
IPI926	CAFs	Orthotopic transplantation tumor in mice	2009	[153]
Halofuginone	CAFs	GEMM	2019	[154]
LIF mAb	CAFs	GEMM	2019	[155]
SOM230	CAFs	Orthotopic transplantation tumor in mice	2016	[156]
Calcipotriene	CAFs	GEMM	2014	[157]
Nintedanib	CAFs	GEMM	2022	[158]
PEGPH20	CAFs	Phase III trial	2020	[159]

## Data Availability

The data presented in this study are available in this article.

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
