# Peer review of "Hydrogel-Based Therapeutics for Pancreatic Ductal Adenocarcinoma Treatment"

_pharmaceutics, 2023, doi:10.3390/pharmaceutics15102421_

Round 1
Reviewer 1 Report
Dear authors:
Thank you for your contribution.
please consider these points in your revision.
1- Please provide a short paragraph concerning using of green polymers in pancreatic cancer therapy.
2- The authors wrote: The release of these cargoes from the hydrogels is mainly affected 63 by factors such as porosity, network expansion/degradation, size of molecules to be re- 64 leased, and drug-polymer interactions (line 63, 64) while they must focus on the environmental factors controlling drug release.
3- It is preferred to supply a list of abbreviation under the abstract.
4- The term hydrogel is being extensively used throughout the manuscript without identifying what type (s) of materials the hydrogel based on.
5- The manuscript is very poor in figures and tables. Organizing the data in tables or representing data by figures are very important in attracting the reader rather than rough text.
6- A section of future perspectives in applying hydrogels in pancreatic cancer therapy should be added.
The manuscript should be revised thoroughly for the grammar and the language.
Author Response
Dear Reviewer
Thank you for giving us the opportunity to submit a revised draft of the manuscript “Hydrogel-based Therapeutics for Pancreatic Ductal Adenocarcinoma Treatment” for publication in the Pharmaceutics. We appreciate the time and effort that you dedicated to providing feedback on our manuscript and are grateful for the insightful comments and valuable improvements to our paper. We have incorporated most of the suggestions. Those changes are highlighted in the manuscript. The point-by-point response to the reviewers’ comments and concerns is also presented in the following section. Besides, we added a co-author “Wenbi Wu” due to her contribution during the revision of this manuscript and we have finished the Authorship Change Form.
Sincerely yours!
Hong Zhu
Response to the reviewer point to point
Comment 1: Please provide a short paragraph concerning using of green polymers in pancreatic cancer therapy.
Response 1: Thanks a lot for your kind suggestion. We have incorporated the application of green polymers (natural polymers) in pancreatic cancer therapy in the hydrogel-based therapeutics for PDAC treatment(From Line 206 on ): “Natural hydrogels are derived of biomaterials found in animal or plant tissues, such as proteins[77], polypeptides[78], polysaccharides[79], and nucleic acids[80]. Gelatin[81], collagen[82], and fibrin[83] are the commonly used natural proteins for hydrogels, while alginate[84], chitosan[85], hyaluronic acid[86], and cellulose[87] are widely used poly-saccharide polymers for hydrogels (Figure 3). Natural polymers are biocompatible, which allows for minimal adverse reactions or immunogenic responses upon implantation or injection[88]. Natural polymers are also biodegradable and can be cleared from the body over time, reducing the risk of long-term complications[88]. Natural hydrogels have been widely used in drug delivery applications for PDAC treatment. They undergo enzymatic degradation by proteolytic enzymes present in the body, allowing for their controlled biodegradation over time[89]. Currently, several natural polymers have been approved by the FDA for application in vivo[90,91]”.
Comment 2: The authors wrote: The release of these cargoes from the hydrogels is mainly affected 63 by factors such as porosity, network expansion/degradation, size of molecules to be re- 64 leased, and drug-polymer interactions (Line 63, 64) while they must focus on the environmental factors controlling drug release.
Response 2: We are grateful for your advice. We have corrected the depiction in this section (Line 65) and discussed the environmental factors influencing the release of drugs in hydrogel (Line 256).
Line 65: “The release of these cargoes from the hydrogels is mainly affected by factors such as porosity, network expansion/degradation, size of molecules to be released, drug-polymer interactions and the environmental stimuli like pH, temperature, enzyme, etc.[22].”
Line 256: “As a particular drug delivery system, hydrogel network can protect drugs from hydrolysis, inactivation and enzymatic hydrolysis by impeding the rapid penetration of various enzymes and provide a prolonged drug release[99,100]. In hydrogels, drug release typically occurs through multiple mechanisms, encompassing diffusion from the porous network[101], degradation of the network[102], hydrogel swelling[103], and deformation of the network[104]. ”
Comment 3: It is preferred to supply a list of abbreviation under the abstract.
Response 3: We appreciate the valuable feedback provided by you. We have supplied the abbreviation list in the revised manuscript and put it after the discussion section according to the journal’s request.
Comment 4: The term hydrogel is being extensively used throughout the manuscript without identifying what type (s) of materials the hydrogel based on.
Response 4: We appreciate the constructive suggestion raised by the reviewer. We have added a list of the hydrogel-based therapeutics for PDAC treatment (Table 1) and identified the specific type of materials the hydrogel is based on.
Comment 5: The manuscript is very poor in figures and tables. Organizing the data in tables or representing data by figures are very important in attracting the reader rather than rough text.
Response 5: We feel regretful for this mistake and great thanks for your professional review work on our article. We have supplied 6 figures and 2 tables in the revised manuscript to better display our manuscript.
Comment 6: A section of future perspectives in applying hydrogels in pancreatic cancer therapy should be added.
Response 6: Thank you for the detailed review. We have added a section of perspectives on applying hydrogels in PDAC in the discussion section of the revised manuscript (From Line 707 on).

Reviewer 2 Report
The manuscript proposed by Jinlu et al for Pharmaceutisc-MPDI is a review reporting about the potential therapeutic use of hydrogels for pancreatic ductal adenocarcinoma treatment. Different examples of hydrogels are described. Even if the therapeutic target is specific, the scope of the paper is in line with the selected journal. The bibliography must be enlarged. I suggest a major revision step according to following point (not in list of importance):
1) I suggest to delate from the title the portion: “ the past, present, and future”.
2) Please, check the formatting (spaces, justification) in all the manuscript (e.g. line 29, line 39).
3) To enlarge the scenario about hydrogels formulation, report also some examples of protein and peptide-based hydrogels (e.g. 10.3389/fchem.2021.770102, 10.3390/gels8120831, 10.1002/chem.201602624). Additionally, include some peptide-based hydrogels for antitumoral drugs delivery.
4) The diagnostic approach to the selected pathology may be included.
5) The manuscript is lacking of Figure. I strongly suggest the inclusion of a Figure about the described pathway involved in the pathology. Its description should be reduced, too.
6) Please, include the different strategies of drugs inclusion in the hydrogels and their possible release mechanism.
7) Include the chemical structure and the reaction for GelMa, as for the tripeptide.
8) Line 299. Include additional examples of nanogels prepared for diagnostic and drug delivery applications, as 10.1002/wnan.1328
9) Line 341. Include the chemical structures.
1) Line 351. Include other examples of peptide-based and polymer-based nanosystems for the delivery of Dox.
1) Line 447. Express PEG
1) Include representative pictures for support the discussion for all the manuscript.
The English quality may be improved.
Author Response
Dear Reviewer
Thank you for allowing us to submit a revised draft of the manuscript “Hydrogel-based Therapeutics for Pancreatic Ductal Adenocarcinoma Treatment” for publication in the Pharmaceutics. We appreciate the time and effort that you dedicated to providing feedback on our manuscript and are grateful for the insightful comments and valuable improvements to our paper. We have incorporated most of the suggestions. Those changes are highlighted in the manuscript. The point-by-point response to the reviewers’ comments and concerns is also presented in the following section. Besides, we added a co-author “Wenbi Wu” due to her contribution during the revision of this manuscript and we have finished the Authorship Change Form.
Sincerely yours!
Hong Zhu
Response to the reviewer point to point
Comment 1: I suggest to delete from the title the portion: “ the past, present, and future”.
Response 1: Thank you for your nice advice on our article. We have corrected the title of this review to “Hydrogel-based Therapeutics for Pancreatic Ductal Adenocarcinoma Treatment”.
Comment 2: Please, check the formatting (spaces, justification) in all the manuscript (e.g. line 29, line 39).
Response 2: We feel regretful for this and thank you so much for your careful review of our manuscript. We have gone through the whole manuscript and revised the format.
Comment 3: To enlarge the scenario about hydrogel formulation, the report also some examples of protein and peptide-based hydrogels (e.g. 10.3389/fchem.2021.770102, 10.3390/gels8120831, 10.1002/chem.201602624. Additionally, include some peptide-based hydrogels for antitumoral drug delivery.
Response 3: We feel great thanks for your professional review work and kind suggestion on our article. We have added the summary of hydrogel formulation in Table 1 and reported the protein and peptide hydrogel (Line 211) in the revised manuscript and given examples of peptide-based hydrogels for pancreatic cancer therapy (from Line 385 on).
Comment 4: The diagnostic approach to the selected pathology may be included. The manuscript is lacking of Figure. I strongly suggest the inclusion of a Figure about the described pathway involved in the pathology. Its description should be reduced, too.
Response 4: Thank you for your kind advice. We have added a figure about the described pathology in PDAC (Figure 2) and reduced the corresponding description.
Comment 5: Please, include the different strategies of drugs inclusion in the hydrogels and their possible release mechanism.
Response 5: We sincerely appreciate the valuable comments. We have summarized the strategies of drug inclusion in the hydrogels and their possible release mechanism in the section “3. Hydrogel-based therapeutics for PDAC” (From Line 245 on).
Comment 6: Include the chemical structure and the reaction for GelMa, as for the tripeptide.
Response 6: Thanks for your nice comments. We have added the chemical structure of GelMA and tripeptide using Figure 3 and Figure 5, respectively.
Comment 7: Line 299. Include additional examples of nano gels prepared for diagnostic and drug delivery applications, as 10.1002/wnan.1328
Response 7: Thanks again for the constructive suggestion. We have included nanogel examples in the revised manuscript and explain the concept and application of nanogels in cancer drug delivery (Lines 232 and 239).
Comment 8: Line 341. Include the chemical structures
Response 8: We appreciate the kindly advice of the reviewer. We have added a figure to display the chemical structures of representative polymers for hydrogel (Figure 3).
Comment 9: Line 351. Include other examples of peptide-based and polymer-based nanosystems for the delivery of Dox
Response 9: Thanks for your suggestion. We have researched the related literature thoroughly. We have not found any study about peptide-based and polymer-based nanosystems for Dox delivery in PDAC treatment. However, some papers reported hydrogel-based delivery systems of DOX for the treatment of other cancers, such as lung cancer, breast cancer, osteosarcoma, and gastric cancer[1-4], which is of guiding significance for hydrogel-based treatment of PDAC. We have added the DOX-loaded nanosystems in the future perspectives of pancreatic cancer therapy (Line 741).
Reference
[1] Y. Qi, H. Min, A. Mujeeb, et al., Acs Applied Materials & Interfaces, 10 (2018) 6972-6981.
[2] W.-H. Chen, W.-C. Liao, Y.S. Sohn, et al., Adv. Funct. Mater., 28 (2018).
[3] M. Zhou, S. Liu, Y. Jiang, et al., Adv. Funct. Mater., 25 (2015) 4730-4739.
[4] C. Chittasupho, J. Angklomklew, T. Thongnopkoon, et al., Polymers, 13 (2021).
Comment 10: Line 447. Express PEG
Response 10: We apologize for this mistake. We have added the expression of PEG in the revised manuscript (Line 528 polyethylene glycol).
Comment 11: Include representative pictures for support the discussion for all the manuscript
Response 11: Thank you very much for your kind advice. We have included 6 pictures in the revised manuscript.

Reviewer 3 Report
I have gone through the review entitled "Hydrogels-based therapeutics for pancreatic ductal adenocarcinoma treatment: the past, present, and future".
This article seems interesting in terms of the application of drug delivery system (particularly, hydrogel materials) for the treatment of pancreatic ductal adenocarcinoma (PDAC) disease. As well, new therapies can be designed to improve future treatments.
The following points need to be addressed:
1) Authors should specify the main objectives in the abstract to capture readers' attention.
2) Correct `line 76`, factors_(Figure 1)
3) Correct `line 15`, Macrophages_(TAMs)
4) Line 235: “Hydrogels can also be designed to degrade and be cleared from the body over time, reducing the risk of long-term complications…” (some examples must be indicated).
5) Lines 240: It is mandatory to specify the hydrodynamic diameters of the drugs (and which drugs). Also, which is the hydrogel pore size (and which hydrogels) and how the pore size can be obtained? These are important facts, and it is a review articles, so the authors need to improve the manuscript.
6) In general, the chemistry of hydrogel materials is not correctly presented, some representative chemical structures can be shown throughout the manuscript. Furthermore, some specific information is missing in the manuscript for a relevant scientific review.
Author Response
Dear Reviewer
Thank you for allowing us to submit a revised draft of the manuscript “Hydrogel-based Therapeutics for Pancreatic Ductal Adenocarcinoma Treatment” for publication in the Pharmaceutics. We appreciate the time and effort that you dedicated to providing feedback on our manuscript and are grateful for the insightful comments and valuable improvements to our paper. We have incorporated most of the suggestions. Those changes are highlighted in the manuscript. The point-by-point response to the reviewers’ comments and concerns is also presented in the following section. Besides, we added a co-author “Wenbi Wu” due to her contribution during the revision of this manuscript and we have finished the Authorship Change Form.
Sincerely yours!
Hong Zhu
Response to the reviewer point to point
Comment 1: Authors should specify the main objectives in the abstract to capture readers' attention.
Response 1: We really appreciate your constructive suggestion. We have revised the abstract.
Comment 2: Correct `line 76`, factors_(Figure 1)
Response 2: Thank you for your valuable comments. We have checked the whole manuscript and corrected this mistake.
Comment 3: Correct `line 15`, Macrophages_(TAMs).
Response 3: Thank you for your valuable comments. We have checked the whole manuscript and corrected this mistake.
Comment 4: Line 235: “Hydrogels can also be designed to degrade and be cleared from the body over time, reducing the risk of long-term complications…” (some examples must be indicated).
Response 4: Thanks for your suggestions. We have rewritten the part of the hydrogel-based drug delivery system and supplemented the classification and monomers of hydrogels. We have indicated that natural hydrogels derived from natural polymers, such as proteins, polypeptides, polysaccharides, and nucleic acids, are biodegradable and can be cleared from the body over time, reducing the risk of long-term complications. Gelatin hydrogels have been widely used in drug delivery applications for PDAC treatment. They undergo enzymatic degradation by proteolytic enzymes present in the body, allowing for their controlled biodegradation over time[92] (Line 219).
Comment 5: Lines 240: It is mandatory to specify the hydrodynamic diameters of the drugs (and which drugs). Also, which is the hydrogel pore size (and which hydrogels) and how the pore size can be obtained? These are important facts, and it is a review articles, so the authors need to improve the manuscript.
Response 5: Thanks for your valuable suggestions. We have rewritten the part of the hydrogel-based drug delivery system and corrected the “hydrodynamic diameters of the drugs” as drug size (Line 261). We have also supplemented the specific information of drug size and hydrogel pore size, as “The hydrogel mesh size represents the dimension of the open space within the porous network. Correspondingly, drug size pertains to the physical dimensions of the drug encapsulated within the hydrogel, including small molecule drugs, proteins, nucleic acids, or cells.”
Comment 6: In general, the chemistry of hydrogel materials is not correctly presented, some representative chemical structures can be shown throughout the manuscript. Furthermore, some specific information is missing in the manuscript for a relevant scientific review.
Response 6: We feel regretful about this. We have checked the expression of the chemistry of hydrogel materials and corrected them. We also show some representative chemical structures in Figure 3. We have checked the whole manuscript and supplemented some specific information in the revised manuscript.

Reviewer 4 Report
1. Authors must update the Global data of pancreatic cancer. They have only reported the USA data.
2. Give a Table for the Hydrogel for different routes of administration for cancer.
3. Give the mechanism of Hydrogel activity on the cancer.
4. Give the Figure for drug loaded hydrogel acting on the cancer cells.
5. Summary of drug used as hydrogel for cancer in Table.
Future prospective and patent section must be added.
Author Response
Comment 1: Authors must update the Global data of pancreatic cancer. They have only reported the USA data.
Response 1: Thanks again for the constructive Comment. We have updated the global data of PDAC (Line 31).
Comment 2: Give a Table for the Hydrogel for different routes of administration for cancer.
Response 2: Thank you very much for the insightful Comments. We have added a table to summarize the different routes of administration of hydrogel-based therapeutics for PDAC treatment (Table 1).
Comment 3: Give the mechanism of Hydrogel activity on the cancer.
Response 3: We appreciate the constructive suggestion raised by the reviewer. We have added a figure and a table to summarize the mechanism of hydrogel activity on cancer (Figure 2 and Table 1).
Comment 4: Give the Figure for drug loaded hydrogel acting on the cancer cells.
Response 4: We appreciate the valuable feedback provided by you. We have added a figure to summarize the effect of hydrogel-based therapeutics on cancer cells and other stroma cells (Figure 2) and a table to summarize the effect, too (Table 1).
Comment 5: Summary of drug used as hydrogel for cancer in Table
Response 5: Thanks a lot for your kind suggestion. We have added a table to summarize the hydrogel-based therapeutics for PDAC treatment (Table 1).
Comment 6: Future prospective and patent section must be added
Response 6: We are grateful for your advice. We have added the section on further perspective in the discussion section.

Round 2
Reviewer 1 Report
The manuscript became highly modified and can be accepted for publication in its from
Reviewer 2 Report
Dear Authors,
I would like to thank you for the effort for the revision step.
In my opinion the manuscript is now suitable for publication.
Reviewer 3 Report
All concerns have been answered. The scientific level of the review has been increased. The paper can now be accepted for publication.
Reviewer 4 Report
Accept